# A Phase-Sensitive Optical Time Domain Reflectometry with Non-Uniform Frequency Multiplexed NLFM Pulse

**DOI:** 10.3390/s23208612

**Published:** 2023-10-20

**Authors:** Zhengyang Li, Yangan Zhang, Xueguang Yuan, Zhenyu Xiao, Yuan Zhang, Yongqing Huang

**Affiliations:** School of Electronic Engineering, Beijing University of Posts and Telecommunications, Beijing 100876, China; lizhengyang@bupt.edu.cn (Z.L.); yuanxg@bupt.edu.cn (X.Y.); zyxiao@bupt.edu.cn (Z.X.); zhang_yuan@bupt.edu.cn (Y.Z.); yqhuang@bupt.edu.cn (Y.H.)

**Keywords:** fiber optic sensors, phase-sensitive optical time domain reflectometer, frequency division multiplexing, non-linear frequency modulation, vibration sensing

## Abstract

In the domain of optical fiber distributed acoustic sensing, the persistent challenge of extending sensing distances while concurrently improving spatial resolution and frequency response range has been a complex endeavor. The amalgamation of pulse compression and frequency division multiplexing methodologies has provided certain advantages. Nevertheless, this approach is accompanied by the drawback of significant bandwidth utilization and amplified hardware investments. This study introduces an innovative distributed optical fiber acoustic sensing system aimed at optimizing the efficient utilization of spectral resources by combining compressed pulses and frequency division multiplexing. The system continuously injects non-linear frequency modulation detection pulses spanning various frequency ranges. The incorporation of non-uniform frequency division multiplexing augments the vibration frequency response spectrum. Additionally, nonlinear frequency modulation adeptly reduces crosstalk and enhances sidelobe suppression, all while maintaining a favorable signal-to-noise ratio. Consequently, this methodology substantially advances the spatial resolution of the sensing system. Experimental validation encompassed the multiplexing of eight frequencies within a 120 MHz bandwidth. The results illustrate a spatial resolution of approximately 5 m and an expanded frequency response range extending from 1 to 20 kHz across a 16.3 km optical fiber. This achievement not only enhances spectral resource utilization but also reduces hardware costs, making the system even more suitable for practical engineering applications.

## 1. Introduction

Optical fiber sensing technology has continuously evolved and improved, surpassing various traditional point monitoring sensor technologies. Among these advancements, distributed optical fiber sensing (DOFS) technology has emerged as particularly noteworthy. DOFS relies on laser-induced backscatter phenomena within optical fibers to monitor physical quantities. This technology has showcased remarkable advantages, including extended monitoring ranges, adaptability to diverse environments, reduced transmission losses, and enhanced system robustness [1,2].

A remarkable advancement within the realm of DOFS is the phase-sensitive optical time domain reflectometer (φ-OTDR). This technology has paved the way for advanced applications like distributed acoustic sensing (DAS). DAS has garnered substantial attention in various applications, notably in architectural health monitoring and security defense. These applications encompass a wide range, from ensuring the structural integrity of bridges and tunnels to safeguarding perimeters [3,4], monitoring pipeline safety [5,6], and providing real-time surveillance for railways and highways [7,8]. The appeal and competitiveness of these sensing systems primarily stem from their exceptional capacity for comprehensive distributed sensing, further complemented by the robustness exhibited by fiber optic sensors when operating in challenging environmental conditions.

Within the context of these applications, critical parameters include measurement distance, spatial resolution, and frequency response range. Current research efforts are dedicated to the simultaneous enhancement of these foundational system attributes. Currently, research on φ-OTDR is mainly focused on four aspects: extending sensing distance, improving spatial resolution, expanding frequency response range, and accurately identifying disturbances [9,10,11,12]. Among these, research on improving spatial resolution mainly centers on the use of pulse compression techniques [13,14,15,16]. Research on expanding the frequency response range primarily concentrates on Frequency Division Multiplexing (FDM) [17,18], fused interferometry [19,20], and periodic non-uniform sampling [21,22].

Combining pulse compression and FDM will occupy a larger bandwidth [23]. The bandwidth is limited by various factors, such as acoustic-optic modulator (AOM) frequency shift and modulation bandwidth, electro-optic modulator (EOM) double-sideband modulation, balanced photodetector (BPD) bandwidth, and acquisition card sampling rate. Current research uses FDM to multiplex the frequency-swept pulses, and each multiplexed frequency-swept pulse occupies the same bandwidth [24,25]. J. Xiong et al. combined positive and negative pulses, FDM technology, and chirped pulse Φ-OTDR, using a 4-frequency band FDM approach that increased the frequency response range by a factor of eight. In a sensing range of 75 km, they achieved a repetition rate of 9.7 kHz, utilizing a total bandwidth of 495 MHz [26]. J. Jiang et al. introduced a Continuous Chirped Pulse Φ-OTDR, achieving a spatial resolution of 4.4 m and a sensing bandwidth of 1.042 MHz on a 1013 m long optical fiber. This approach utilized a total bandwidth of 1.6 GHz [27]. It was found that a large number of frequency points are required for traditional FDM to achieve wide frequency response, especially when high spatial resolution is also desired. Each frequency point needs a wide bandwidth, which consumes a significant amount of spectrum resources and imposes high bandwidth requirements on the modulator and detector. Therefore, a non-uniform FDM approach is proposed, which can achieve wide frequency response and high spatial resolution while increasing the utilization of spectrum resources. It also allows for more frequency multiplexing within the same bandwidth as traditional FDM, thus expanding the frequency response range.

In this paper, a non-uniform frequency division multiplexing with a nonlinear frequency modulation (NUFDM-NLFM) approach that combines pulse compression and non-uniform FDM is proposed. Frequency-swept pulses with different bandwidths are multiplexed. Experimental validation was conducted by multiplexing eight frequencies within a 120 MHz bandwidth. The findings demonstrate a spatial resolution of approximately 5 m and an extended frequency response range spanning from 1 to 20 kHz over a 16.3 km optical fiber. This approach allows for the achievement of higher spatial resolution and a broader frequency response range while enhancing spectral resource utilization and conserving hardware resources, thereby providing a solid foundation for the practical engineering application of this system.

## 2. Principle and Theoretical Analysis

The principle diagram of φ-OTDR using NUFDM-NLFM is shown in Figure 1. The light emitted by the light source is divided into the probe light and the local oscillator light by a coupler. The probe light is modulated by a modulator to generate a chirped pulse and then transmitted to the fiber under test (FUT) through an optical circulator (OC). The Rayleigh backscattering (RBS) light from the FUT beats with the local oscillator light. The detected optical signal is converted into an electrical signal and then collected by a data acquisition card (DAQ).

The frequency-swept pulse with a large bandwidth can improve spatial resolution. The frequency-swept pulse with a small bandwidth can carry the phase information to improve the frequency response range of vibration. Simultaneously using frequency-swept pulses with different bandwidths can reduce the required spectrum resources and lower the hardware requirements of the sensing system.

Modulation signals in the non-uniform FDM-NLFM φ-OTDR system are composed of multiple NLFM signals [28,29]. A single frequency-swept pulse can be expressed as
(1)$$E_{p}=rect(t/T)\exp\left[j2\pi f_{c}t+j2\pi f_{0}t+j\theta \left(t\right)\right]$$
where rect(t/T) is the rectangular window function for t∈[0,T], T is the pulse width, $f_{c}$ is the optical frequency, $f_{0}$ is the center frequency of the frequency-swept pulse electrical signal, and θ(t) is the phase function of the frequency-swept pulse.

Backward Rayleigh scattered light and local oscillator light are detected coherently, and the signal is converted to an electrical signal $I_{beat}$. It is necessary to perform matched filtering on $I_{beat}$ to obtain a compressed pulse. The matched filter can be represented as
(2)h=exp[−jθ(T−t)]

$I_{beat}$ is down-converted to remove the center frequency $f_{0}$, and then convolved with the matched filter h to obtain the compressed pulse S, which can be expressed as
(3)$$S=conv\left(I_{beat}\exp\left(-j2\pi f_{0}t\right),h\right)$$

S is determined by the phase function of the frequency-swept pulse and the phase information in the fiber. The spatial resolution R is only related to the bandwidth B occupied by the frequency-swept pulse and is independent of the pulse width T. When T is increased to improve the sensing distance, a higher spatial resolution can be obtained simultaneously by matched filtering.

Crosstalk caused by the side lobes of the compressed pulse needs to be reduced by increasing the side lobe suppression ratio [30]. Due to the narrow main lobe width and low side lobe suppression ratio of the linear frequency modulation (LFM) signal, the windowing method in the time domain is usually used to improve the side lobe suppression ratio of the LFM signal [31,32]. However, windowing causes the main lobe to widen, reducing spatial resolution. In contrast, the NLFM method features lower sidelobes and a well-maintained main lobe width [33,34]. Therefore, NLFM is used to improve the side lobe suppression ratio [35,36].

The inverse synthesis design of NLFM signals using a window function based on the stationary phase principle was adopted [37,38]. The NLFM signal is represented as
(4)x=exp[jθ(t)]

In the equation, θ(t) represents the phase function of the signal, the Fourier transform of x is denoted as X(f), and the matching filter for x is given by the Formula (2) as h, with the Fourier transform of h represented as H(f).

The spectrum of the matched filter’s output, Y(f), is given by:(5)$$Y(f)=X(f)H(f)={\left|X(f)\right|}^{2}$$

To enhance the sidelobe suppression ratio, time domain windowing techniques are often employed. The Hanning window is utilized due to its favorable attribute of maintaining a narrow main lobe width within the frequency domain [39]. In the process of generating NLFM signals, the Hanning window was used. For a certain frequency domain Hanning window function W(f), let
(6)$$\begin{array}{c} {\left|X\left(f\right)\right|}^{2}=W\left(f\right) \end{array}$$

The relationship between signal spectrum and frequency modulation slope can be expressed by the stationary phase principle as follows
(7)$$\begin{array}{c} X\left(f\right)\propto \sqrt{\frac{1}{\theta^{\prime \prime }\left(t\right)}}=\sqrt{\frac{1}{\frac{df\left(t\right)}{dt}}}=\sqrt{\frac{dT_{g}\left(f\right)}{df}} \end{array}$$
where $T_{g}(f)$ is the group delay function of the signal, f(t) is the frequency modulation function, and they are inverse functions of each other W(f) is proportional to the derivative of $T_{g}(f)$ with respect to f. Therefore, the group delay function $T_{g}(f)$ can be obtained by integrating W(f).
(8)$$\begin{array}{c} T_{g}\left(f\right)=\frac{T}{\int_{-\frac{B}{2}}^{\frac{B}{2}}W\left(v\right)dv}\int_{-\infty }^{f}W\left(v\right)dv \end{array}$$

In the equation, K is a constant coefficient. When the NLFM signal is required to have a frequency chirping range of B and a pulse width of T, it can be expressed as:(9)$$K=\frac{T}{\int_{-\frac{B}{2}}^{\frac{B}{2}}W(v)dv}$$

According to the relationship between $T_{g}(f)$ and f(t), the following can be obtained
(10)$$\begin{array}{c} f\left(t\right)=T_{g}^{-1}\left(f\right) \end{array}$$

Therefore, the phase function of the signal is given by
(11)$$\begin{array}{c} \theta \left(t\right)=2\pi \int_{-\infty }^{t}f\left(\tau \right)d\tau \end{array}$$

The compressed pulse S obtained through matched filtering is a complex number, and taking the modulus of S yields the amplitude abs(S) containing the vibration information. Noise can be reduced by performing moving average smoothing on the amplitude and then performing the moving differential to determine the vibration position [40]. However, the relationship between amplitude and vibration waveform is nonlinear, and to restore the accurate vibration waveform, it is necessary to take the principal value of the phase angle(S), thus obtaining the phase of the compressed pulse. The relationship between phase and vibration waveform is linear, and phase unwrapping is required during phase demodulation. Phase unwrapping can make the phase changes of demodulation continuous and restore the real vibration waveform [41].

When a single-frequency pulse enters the sensing optical fiber with a pulse period of $T_{p}$ and a length of L, it must satisfy $T_{p}\geq 2nL/c$ to ensure that the backscattered Rayleigh scattering signals corresponding to different pulses do not overlap. The pulse period $T_{p}$ and the sampling rate $f_{s}$ satisfy $f_{s}=1/T_{p}$, meaning $f_{s}<c/2nL$, indicating that for longer sensing distances, the sampling rate becomes lower. However, Frequency Division Multiplexing (FDM) can overcome this limitation by using pulse signals in different frequency bands, avoiding mutual interference between the backscattered signals corresponding to each pulse, and making the pulses more densely packed, thus increasing the sampling rate $f_{s}$. FDM requires the phase demodulation of each frequency band to be completed first and then concatenates the phase demodulation results of different frequency bands in time in order to ultimately increase the sampling rate.

When using a frequency-swept pulse, the sweep range determines the spatial resolution of using amplitude positioning. Typically, the frequency bandwidth of each frequency band used by FDM is the same, but when FDM is combined with frequency-swept pulses, all frequency bands do not need to have the same spatial resolution. Phase demodulation can be performed solely in the phase-stable region, and the accuracy of the restored vibration waveform is preserved. Therefore, non-uniform FDM can be used to save spectrum resources. A large sweep range pulse is used to improve spatial resolution and achieve accurate positioning. Multiple small sweep range pulses are used to carry phase variation information to improve the system sampling rate. The fiber-optic distributed acoustic sensing system designed in this paper can save spectrum resources and reduce hardware requirements compared to existing systems that combine frequency-swept pulses with FDM.

## 3. Experiment and Result

The experimental setup is configured as shown in Figure 2. The laser produced by the narrow linewidth laser (SFFL-N-34) is split into local oscillator light and signal light by coupler OC1. The signal light is modulated into a non-uniform FDM-NLFM signal by EOM (10 G LN Optical Modulator), and AOM (T-M080-0.4C2J-3-F2S) chops the signal to improve the extinction ratio. The pulsed light is amplified by an Erbium-doped fiber amplifier (EDFA), and the amplifier spontaneous emission (ASE) noise is filtered out by fiber Bragg grating (FBG). The signal light enters the approximately 16.3 km long optical fiber. Two piezoelectric transducers (PZT) are sequentially placed at the end of the FUT with a 5-m interval. The PZT coiled 5 m optical fiber is labeled as PZT1, while the PZT coiled 10 m optical fiber is designated as PZT2. The backscattered signal from the fiber and the local oscillator light enter coupler OC2, are detected by BPD, and converted into electrical signals.

The electrical signal passes through a high-pass filter to eliminate unnecessary spectral components before mixing to prevent overlap. It then enters a mixer, where it is mixed with an 80 MHz sine wave to eliminate the 80 MHz frequency shift introduced by the AOM. The mixed signal goes through a low-pass filter to remove mirror frequencies while reducing the power spectral density of noise, thereby improving the signal-to-noise ratio. Subsequently, the signal enters an acquisition card with a sampling rate of 250 MSa/s and is uploaded to a PC. Through the PC, each frequency band is down-converted separately, passed through a low-pass filter with a bandwidth corresponding to the frequency band, and then subjected to matched filtering. This process extracts the amplitude for positioning and simultaneously extracts the phase to reconstruct the vibration waveform.

### 3.1. High-Frequency Vibration Sensing Experiment

The experimental parameters designed in this study are: a spatial resolution of about 5 m, a sensing distance of about 16.3 km, and a frequency response range of 1–20 kHz.

When using a frequency-swept pulse, the sweep range determines the spatial resolution when using amplitude positioning. However, when FDM is combined with a frequency-swept pulse, it is not necessary for all the frequency bands of a frequency-swept pulse to have the same spatial resolution for amplitude-based positioning. The determination of vibration positions and ranges is governed by signal intensity and does not require phase demodulation or the merging of multiple signals. As long as one signal among the multiple signals can accurately determine the vibration positions and ranges through intensity, it suffices. Signals with larger bandwidths provide the best achievable spatial resolution, so the spatial resolution for positioning is determined solely by signals with larger bandwidths. Even if spatial resolutions differ, as long as phase demodulation is performed in regions with phase stability, the reconstructed vibration waveform remains accurate.

Once vibration positions are determined, phase demodulation is obtained from the time-series signals at specific positions, allowing for the independent extraction of the correct phase changes for each vibration point. Specifically, the spatial resolution for positioning is entirely determined by signals with a larger bandwidth. Even if signals with a smaller bandwidth offer a spatial resolution greater than 5 m, when synthesizing the phase results, two vibration points separated by 5 m can still be better distinguished using signals with a larger bandwidth. Signals with a larger bandwidth are employed to establish separation points between two vibration positions. Phase demodulation is performed both before and after these separation points, enabling the separate and accurate extraction of the phase changes associated with each vibration point, even if signals with a smaller bandwidth provide a spatial resolution greater than 5 m.

Therefore, the system designed in this paper uses signals with larger bandwidths for positioning and determining spatial resolution, while multiple signals with smaller bandwidths enhance the frequency response range.

As illustrated in Figure 3, the approach entails utilizing the leftmost narrow-peaked, high-bandwidth frequency-swept signal for positioning and enhancing the system’s spatial resolution. Meanwhile, multiple broader-peaked, low-bandwidth frequency-swept signals are employed to extend the system’s frequency response range.

Therefore, the NLFM pulse has a scanning range of *B* = 30 MHz, a pulse width of *T* = 4 μs, a sampling rate of *f_s_
*= 5 kHz for each NLFM pulse, a pulse period of *T_p_
*= 200 μs, and eight different frequency ranges of NLFM pulse are multiplexed. In addition to the necessary NLFM pulse with a scanning range of 30 MHz, there are still 90 MHz available frequency bands that require the multiplexing of seven NLFM pulses, each with a scanning range of 12 MHz and a pulse width of 4 μs. FFT results of the designed pulse sequence are shown in Figure 4a.

In the experiment, a sinusoidal wave with a frequency of 19.8 kHz and a peak-to-peak voltage of 1 Vpp was applied to PZT1 located at 16.3 km; the power spectrum of the signal received by the acquisition card is shown in Figure 4b.

Matched filtering was applied to the 30 MHz NLFM signal used for positioning, and the complex modulus abs (S) obtained from the matching filter was subjected to moving average filtering with a window size of 8 to reduce noise. Subsequently, moving differential filtering was employed to determine the vibration position, as illustrated in Figure 4a. In Figure 5a, a peak appears at 16.3 km, which corresponds to the vibration position of the applied 19.8 kHz sine wave, with a location signal-to-noise ratio of approximately 10 dB, thus demonstrating that the optical fiber sensing system is capable of vibration localization.

Zooming in on the peak of the moving difference along the optical fiber at 16.3 km, the results are shown in Figure 5b. The −3 dB point of the vibration peak in the figure represents the full width at half maximum (FWHM) of the compressed pulse, which represents the spatial resolution of the compressed pulse. As shown in the figure, the spatial resolution is about 5 m, which proves that the sensing system achieves the design requirement for locating vibrations using a 30 MHz NLFM pulse.

In the context of acoustic signal propagation through optical fibers, the utilization of broadband frequency pulses ensures that spatial accuracy along the fiber’s axis remains unproblematic. Meanwhile, in the temporal domain, the presence of precise intervals between distinct points allows for the enhancement of vibrational propagation imaging detail through suitable fitting techniques to address any potential limitations.

We extracted the principal argument angle of the complex numbers obtained by match filtering eight frequency components within the region centered at 16.3 km along the fiber, denoted as $angle(S_{j}),(1\leq j\leq 8)$, and performed phase unwrapping to obtain the restored vibration waveform, as shown in Figure 6a. The vibration signal applied to PZT1 in the experiment is a 19.8 kHz sine wave. Under the undersampling condition, the frequency of a single compressed pulse in the acquired vibration waveform should be 200 Hz. The results obtained by sampling eight compressed pulses, shown in Figure 6a, are all sine waves with a frequency of 200 Hz, which proves that all eight frequency components of the sensing system can accurately capture the vibration signal.

In the experiments conducted in this study, the length used for phase demodulation was consistently 84 m. Concatenating the waveforms obtained by phase demodulation of the eight NLFM pulses in time order, the resulting vibration waveform is shown in Figure 6b. Take the fast Fourier transform of the vibration waveform shown in Figure 6b, and the resulting power spectrum is shown in Figure 6c. There is a peak at the frequency of 19.8 kHz, which corresponds to the vibration signal, and the signal-to-noise ratio is about 30 dB. Figure 6b,c demonstrates that this sensing system can improve the frequency response range of vibration signals through FDM. By concatenating the phase-demodulated results of multiple NLFM pulses, it is possible to accurately reconstruct high-frequency vibration waveforms.

### 3.2. Bandwidth Vibration Sensing Experiment

Under the condition of a peak-to-peak driving voltage of 400 mVpp and a frequency sweeping range of 1~20 kHz, with a rise/fall time of 20 ms, the phase demodulation waveform obtained by the sensing system is shown in Figure 6a.

The result of applying the short-time Fourier transform to the waveform shown in Figure 7a is shown in Figure 7b. This experimental result demonstrates that the frequency response range of this sensing system expands from 1–2.5 kHz to 1–20 kHz through FDM.

### 3.3. Demodulating Signals with Different Bandwidths

Using frequency bands of different widths has no effect on the demodulated signal, as the demodulation involves the accumulated phase changes within the phase-stable region, which remain unrelated to the variation in scattered signal intensity. Additional experiments were conducted to verify the above conclusion. The same 200 Hz, 1 Vpp vibration at the same location was applied, and sweep bandwidths of 30 MHz and 12 MHz were used to demodulate the signals. The demodulated signals are depicted in Figure 8 below.

From the above experimental results, it can be observed that when applying the same 200 Hz, 1 Vpp vibration at the same location, the phase peak-to-peak value of the demodulated signal using a 30 MHz sweep bandwidth is 6.715032785557739 rad, while the phase peak-to-peak value of the demodulated signal using a 12 MHz sweep bandwidth is 6.730464131544842 rad. Thus, it can be concluded that, when using different sweep bandwidths, the demodulated signals of the same vibration or frequency sweep change at the same location are indeed the same.

Furthermore, due to the nonlinear nature of the frequency modulation used in NLFM, even if a frequency band with the same bandwidth as others is extracted, the spatial resolution remains different. If the pulse width of a wideband pulse is increased to be higher than other frequency bands, and the compressed pulse with the same bandwidth is forced to have the same spatial resolution, the spectral power outside the band will be extremely low, thereby unable to effectively improve spatial resolution.

### 3.4. Multi-Position Vibration Sensing Experiment

At a distance of 16.3 km along the tested optical fiber, a PZT1 with a coiled 5 m optical fiber was positioned and subjected to vibration at a frequency of 9.8 kHz and a peak-to-peak driving voltage of 1 Vpp. Another PZT2 with a coiled 10 m optical fiber was placed 5 m away and similarly subjected to a vibration signal at a frequency of 9.8 kHz and a peak-to-peak driving voltage of 1 Vpp. The positioning results obtained through the differential movement of this sensing system are illustrated in Figure 9a. The system successfully identified two vibration positions that were 5 m apart, corresponding to PZT1 and PZT2, respectively. Phase demodulation was performed on the vibrations at these two positions, followed by FFT analysis, resulting in the power spectra depicted in Figure 9b,c.

Considering the comprehensive results of the aforementioned experiments, it can be concluded that the proposed sensing system in this study possesses the capability to identify multiple points of vibration.

### 3.5. Strain Resolution and Sensitivity

In recent years, researchers have delved deeply into the quantitative assessment and enhancement of DAS sensitivity. Fading suppression techniques and laser phase-noise compensation methods stand out as two notable approaches in this regard. In 2018, Luis Costa and colleagues achieved a resolution performance of 5 pε/√Hz @ 1 kHz using Chirped Pulse F-OTDR with phase-noise compensation technology [42]. In 2019, Mengshi Wu and co-workers achieved a resolution of 92.84 pε/√Hz @ 500–2500 Hz through the use of Pulse Compression with phase-noise compensation technology [43].

The distributed optical fiber acoustic sensing system proposed in this study employs a single-mode quartz optical fiber with a core refractive index of 1.46, a propagation constant of 2π/1550.12, and an equivalent elastic-optic constant of 0.22. As shown in Figure 10a, the system’s optical fiber has a PSD baseline noise of −73.3034 dB rad^2^/Hz. Under both the utilization of 30 MHz and 12 MHz scanning signals, the strain resolution of this system is measured to be $0.2813p\varepsilon /\sqrt{\mathrm{Hz}}$.

The relationship between the PZT1 driving voltage and the amplitude of the demodulated phase is shown in Figure 10b; the voltage-strain conversion relationship of PZT1 is 0.2356 μs/V, and the optical fiber distributed acoustic sensor proposed in this paper has good linear strain response with a strain sensitivity of 2.716 rad/(με·m) and $R^{2}=0.9999$.

## 4. Conclusions

The experiment used a frequency bandwidth of 120 MHz and multiplexed 8 frequencies, achieving a spatial resolution of 5 m and a frequency response range of 1–20 kHz over a 16.3 km fiber. Compared with uniform FDM, non-uniform FDM achieves the same performance, saves up to 50% of the spectrum resources, and greatly reduces hardware requirements.

Considering the comprehensive results of the aforementioned experiments, it can be concluded that the proposed sensing system in this study possesses the capability to identify multiple points of vibration. Using non-uniform FDM can reduce the spectral resources required for the combination of FDM and compressed pulses, and this method does not affect the spatial resolution when using amplitude positioning, nor does it affect the accurate restoration of the vibration waveform when performing phase demodulation.

The cost-effective and resource-efficient nature of non-uniform FDM suggests that it can be a valuable tool for various applications in optical fiber sensing. Further investigations can explore the full extent of its capabilities and potentially lead to enhanced sensing techniques and broader practical implementations.

## Figures and Tables

**Figure 1 sensors-23-08612-f001:**
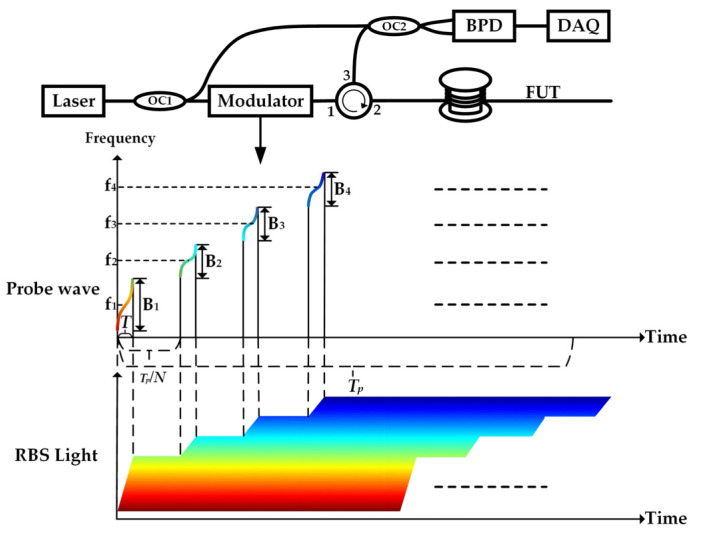
Principle of phase-sensitive optical time domain reflectometer with non-uniform frequency multiplexing of NLFM pulses.

**Figure 2 sensors-23-08612-f002:**
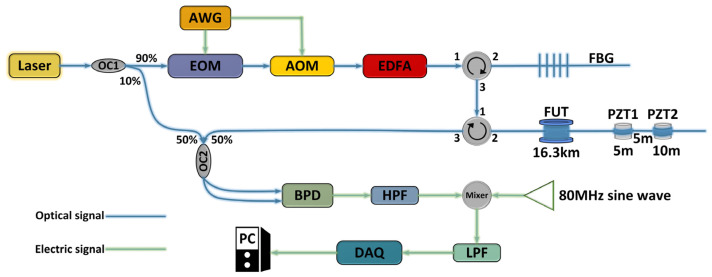
Experimental setup.

**Figure 3 sensors-23-08612-f003:**
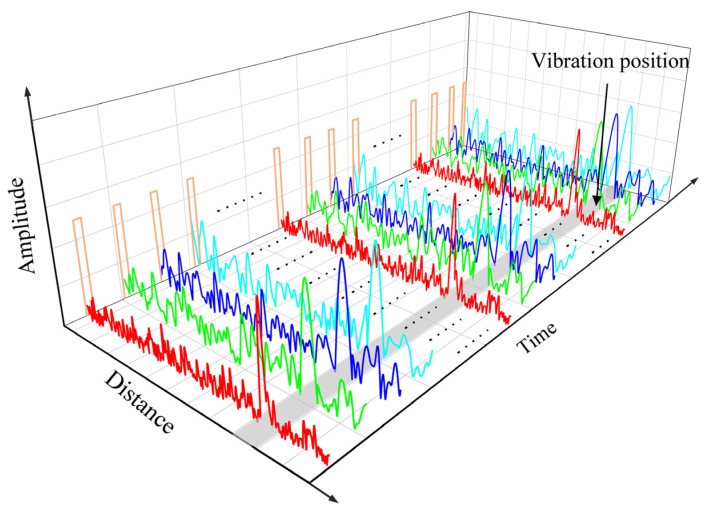
Schematic diagram of non-uniform frequency multiplexed NLFM pulse sampling.

**Figure 4 sensors-23-08612-f004:**
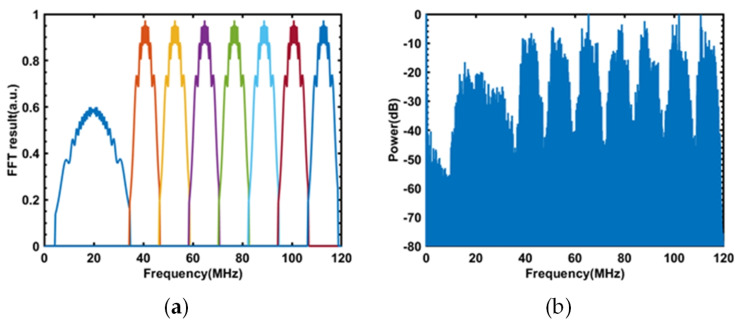
(**a**) FFT results of the designed pulse sequence. (**b**) The frequency spectrum of non-uniform FDM-NLFM.

**Figure 5 sensors-23-08612-f005:**
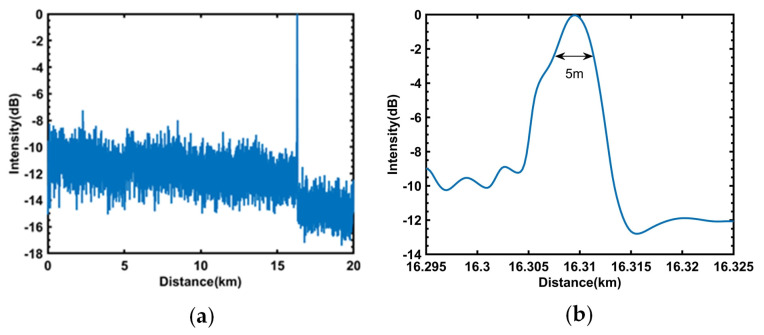
(**a**) The vibration positioning of the 30 MHz NLFM signal. (**b**) The spatial resolution of the 30 MHz NLFM signal.

**Figure 6 sensors-23-08612-f006:**
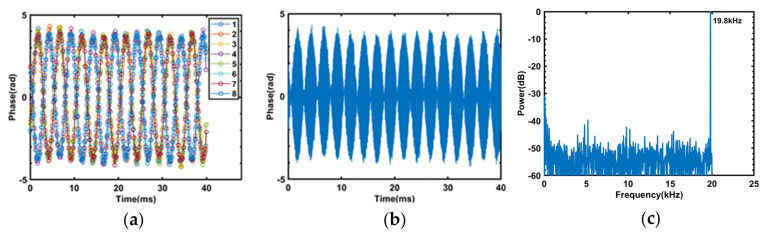
(**a**) The phase demodulation results of a 19.8 kHz 1 Vpp sine wave after undersampling. (**b**) The phase demodulation results of a 19.8 kHz 1 Vpp sine wave. (**c**) The power spectrum of a 19.8 kHz 1 Vpp sine wave.

**Figure 7 sensors-23-08612-f007:**
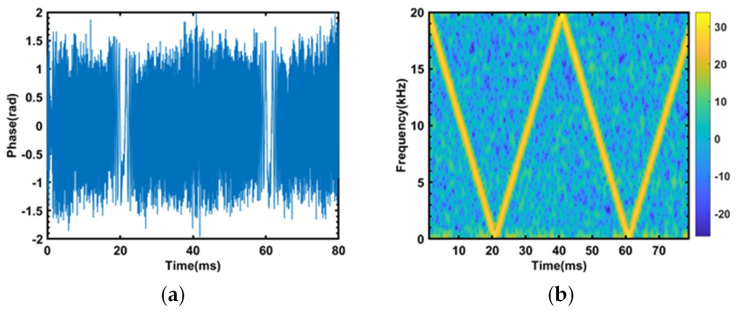
(**a**) The phase demodulation result of a 1~20 kHz 400 mVpp sine wave. (**b**) The short-time Fourier transform result of a 1~20 kHz 400 mVpp sine wave.

**Figure 8 sensors-23-08612-f008:**
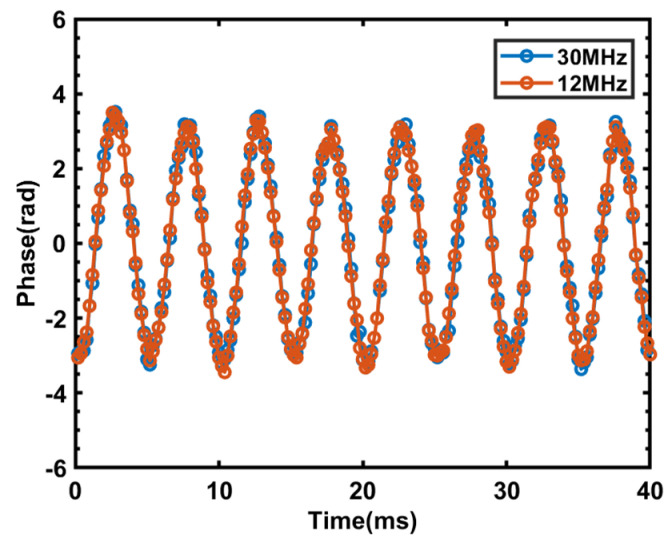
Results of phase demodulation of a 200 Hz, 1 Vpp vibration signal using different sweep bandwidths.

**Figure 9 sensors-23-08612-f009:**
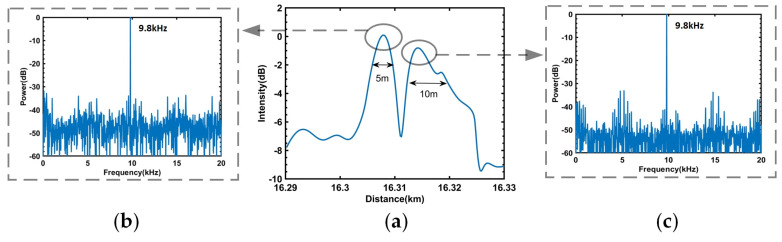
(**a**) Moving differential results. (**b**) The power spectrum of PZT1. (**c**) The power spectrum of PZT2.

**Figure 10 sensors-23-08612-f010:**
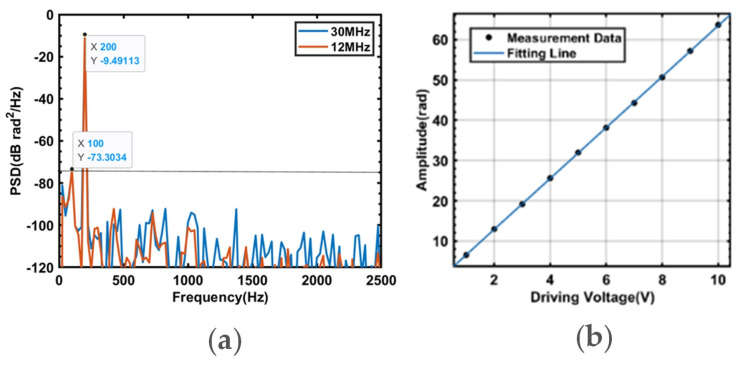
(**a**) Power spectral density plot of a 200 Hz, 1 Vpp vibration signal. (**b**) The relationship between the PZT1 driving voltage and the amplitude of the demodulated phase.

## Data Availability

The data presented in this study are available upon request from the corresponding author.

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
