# Peer review of "A Phase-Sensitive Optical Time Domain Reflectometry with Non-Uniform Frequency Multiplexed NLFM Pulse"

_sensors, 2023, doi:10.3390/s23208612_

Round 1

Reviewer 1 Report

  • Based on the FFT results of the designed pulse sequence shown in Figure 3, the scanning range for 7 frequencies are all 12MHz. Therefore, the final spatial resolution should be larger than 10 meters. Since the spatial resolution of each frequency is greater than 5 meters, it is impossible to achieve a spatial resolution of 5 meters even when multiple frequencies are combined.
  • The article does not specify the length during phase demodulation.
  • "However, windowing also causes the height of the main lobe to decrease, resulting in a loss of signal-to-noise ratio." I believe the understanding of the effects of windowing is incorrect. Windowing does not result in a loss of signal-to-noise ratio. The broadening of the main lobe should lead to a decrease in spatial resolution, but in terms of demodulation signal-to-noise ratio, it should be better.
  • It is impossible to understand the meaning of Figure 4 based on the main text and the figure caption. Moreover, from Figure 4(b), it can be seen that there are many peaks near the vibration point. I highly doubt that the frequency reuse method used in this paper has caused significant crosstalk.
  • Through which figure is the PSD baseline noise of this paper observed?

The author should elaborate in detail on these critical issues, especially concerning the concerns about spatial resolution results. Moreover, the experimental results in this paper are not rigorous enough and require substantial revisions to meet publication standards.

Reviewer 2 Report

The paper aims to combine two different techniques for phase-sensitive distributed acoustic sensing: pulse compression via non linear frequency modulation (NLFM) and non-uniform Frequency Division Multiplexing.

While the methodology of the experiment seems correct and the results seem accurate, I believe there are a few aspects on which the authors could elaborate more

First, the improvement provided by non uniform FDM could benefit from further explanation. It is mentioned in section 2, in lines 83-84, but it is limited to two phrases. A more articulate explanation of how a single larger bandwidth pulse followed by several smaller bandwidth ones can help improve spatial resolution and frequency response might be beneificial in making the benefits of this work clearer.

In section 3, in the description of the measurement setup, it would help to better elucidate how the various high and low pass filters allow to extract the trace for the different pulses.

It would also prove useful in section 3.5 to compare the results in terms of sensitivity and resolution to similar designs developed in previous works.

Other minor issues:

Line 131: there is a question mark that should not be there.

Line 145: "by moving the average on the amplitude" should be changed to "by performing moving average smoothing on the amplitude". In addition, the phrase "Reducing noise by moving the average on the amplitude, and then performing the moving differential to determine the vibration position." is incomplete. For instance, it could be changed into "Noise can be reduced by performing moving average smoothing on the amplitude and then performing the moving differential to determine the vibration position".

Line 153: there is no real link between the two paragraphs. The paragraph above talks about general phase OTDR phase demodulation and unwrapping. The paragraph below talks about FDM and crosstalk, and starts with "however", as if it was a continuation of the previous one, despite being about a different argument.

On Line 206, moving average filtering is mentioned. It could be useful to express the size of the window.

Line 215: "Amplifying the peak of the moving difference" should be changed to "Zooming in on the peak of the moving difference"

Line 226 to 229. The phrase here is not well conjugated. It could be fixed by changing "Extract" at line 226 and "perform" at line 228 with "We extracted" and "performed" respectively.

The English language seems overall solid.

I spotted some mistakes, that were reported in the suggestions for Authors.

Reviewer 3 Report

1. The phase-sensitive optical time domain reflectometers in the second paragraph does not have a strong logical connection with the first paragraph. You are advised to revise it.

2. It is suggested that the author add some new literature of recent three years

Round 2

Reviewer 1 Report

I believe the author did not adequately address the most crucial question. The statement, "The spatial resolution for positioning is determined solely by signals with larger bandwidth," is certainly incorrect. This is because the final phase result synthesizes all the demodulated phases from different frequency bands.
